# Estimation of Reference Evapotranspiration in a Semi-Arid Region of Mexico

**DOI:** 10.3390/s23157007

**Published:** 2023-08-07

**Authors:** Gerardo Delgado-Ramírez, Martín Alejandro Bolaños-González, Abel Quevedo-Nolasco, Adolfo López-Pérez, Juan Estrada-Ávalos

**Affiliations:** 1Hydrosciences, Postgraduate College, Campus Montecillo, México-Texcoco Highway Km 36.5, Montecillo 56264, Mexico; delgado.gerardo@colpos.mx (G.D.-R.); anolasco@colpos.mx (A.Q.-N.); adolfholp@colpos.mx (A.L.-P.); 2National Institute for Forest, Agriculture and Livestock Research (INIFAP), National Center for Disciplinary Research on Water, Soil, Plant and Atmosphere Relationships (CENID-RASPA), Right Bank Sacramento Channel Km 6.5, Gómez Palacio 35150, Mexico; estrada.juan@inifap.gob.mx

**Keywords:** NASA-POWER platform, empirical equations, reanalysis data, meteorological data

## Abstract

Reference evapotranspiration (ET_0_) is the first step in calculating crop irrigation demand, and numerous methods have been proposed to estimate this parameter. FAO-56 Penman–Monteith (PM) is the only standard method for defining and calculating ET_0_. However, it requires radiation, air temperature, atmospheric humidity, and wind speed data, limiting its application in regions where these data are unavailable; therefore, new alternatives are required. This study compared the accuracy of ET_0_ calculated with the Blaney–Criddle (BC) and Hargreaves–Samani (HS) methods versus PM using information from an automated weather station (AWS) and the NASA-POWER platform (NP) for different periods. The information collected corresponds to Module XII of the Lagunera Region Irrigation District 017, a semi-arid region in the North of Mexico. The HS method underestimated the reference evapotranspiration (ET_0_) by 5.5% compared to the PM method considering the total ET_0_ of the study period (26 February to 9 August 2021) and yielded the best fit in the different evaluation periods (daily, 5-day mean, and 5-day cumulative); the latter showed the best values of inferential parameters. The information about maximum and minimum temperatures from the NP platform was suitable for estimating ET_0_ using the HS equation. This data source is a suitable alternative, particularly in semi-arid regions with limited climatological data from weather stations.

## 1. Introduction

Evapotranspiration (ET) is the sum of transpiration through the plant canopy and evaporation from the soil, plant, and free surface water [1,2]. ET is the most significant component of the hydrological cycle [1,3], due to which its estimation is of common interest in climatological, hydrological, forestry, and agricultural studies [4]. This last area ET is a fundamental variable for calculating water requirements, making efficient use of water in crop production [5]. ET can be measured directly using weighing lysimeters or by measuring the net flux of water vapor between the surface and the surrounding atmosphere using micrometeorological methods [6], which depend on the energy balance of the canopy and include the energy balance of Bowen’s relation, eddy covariance, and the use of scintillometers [7].

Crop evapotranspiration (ET_C_) is a crucial aspect of the water balance in agricultural areas. To estimate it, the most accessible method is to estimate reference evapotranspiration (ET_0_) and then pair it with crop and soil coefficients [8]. Reference evapotranspiration (ET_0_) is the evapotranspiration rate of a hypothetical reference crop (grass or alfalfa) with a height of 0.12 m, a fixed surface resistance of 70 s m^−1^, and an albedo of 0.23, homogeneous, well-watered, free from diseases and pests, growing vigorously, and providing complete shade to the soil [9,10,11]. ET_0_ measures atmospheric evaporation demand regardless of crop type, development, and management practices [12,13]. This variable is affected only by climatic factors [14] and can be calculated from meteorological data [10]. 

Estimating ET_0_ is the first step in designing, planning, and managing different irrigation systems [15,16]. In addition, it is relevant for calculating crop water requirements [17,18]. This parameter is the backbone of the agronomic design of any irrigation system, facilitates its operation (irrigation schedule and shifts), and allows the planning of water resource management in a basin [19] or an irrigation district. Therefore, its accurate estimation is essential in water management, particularly in arid and semi-arid areas where water is scarce [20].

Given its importance, and the climate’s temporal and spatial variability, many models to estimate ET_0_ have been proposed. In general, the models available in the published literature can be broadly classified as follows: (1) fully physically based models on a combination of energy balance and mass transfer; (2) semi-physical models based on temperature, radiation, and evaporation data; and (3) black-box models based on artificial neural networks, empirical relationships, and genetic and fuzzy algorithms [21,22]. 

Due to its practicality, many empirical equations have been developed from field experiments and those based on theoretical approaches [19]. These methods include the evaporimeter tank and empirical equations, including the complete physical model (FAO-56 Penman–Monteith), the equation based on temperature (Blaney–Criddle, Thornthwaite, and Turc), and the one based on temperature and radiation (Hargreaves, Jensen–Haise, Priestley–Taylor, and FAO Radiation), among others [21]. 

The UN Food and Agriculture Organization (FAO) recommends the Penman–Monteith standard method described in the FAO-56 Manual because it can be used in arid, temperate, and tropical areas [23]. Furthermore, this standardized method is more accurate than the standard proposed by the American Society of Civil Engineers (ASCE), ASCE-PM, when estimating daily ET_0_; both ways were compared with lysimetric measurements [24]. However, this method requires various meteorological input variables (temperature, solar radiation, relative humidity, and wind speed), which restrains its widespread use [25]. Therefore, its usefulness is limited in regions with no meteorological stations or a shortage of input data [26], which are usually unavailable with the required frequency and quality [27]. The other equations can be used in regions with very little climatological information, such as the case of Hargreaves–Samani (HS) and Blaney–Criddle (BC) equations, which are the most common ones [28,29,30] and only require temperature as an input variable [31]. 

The accuracy of the HS and BC equations has been evaluated by several authors, comparing their results with the FAO-56 Penman–Monteith (PM) reference method; HS was the equation that attained the best fit in semi-arid regions [32,33]. Other authors state that the HS method works well in most climatic regions, except for wet areas where it tends to overestimate ET_0_ [16,34,35,36]. Since HS was developed empirically based on data from arid to subhumid environments, it may not fit well to conditions markedly different from those considered for its calibration, as is the case of wet climates [16]. On the other hand, the HS method underestimates ET_0_ for dry and windy areas because it does not include wind and is seemingly more accurate when applied for 5- to 7-day averages than for daily time scales [37,38]. However, despite a reasonably good performance of the HS equation in most applications, particularly irrigation planning, several authors have attempted to either recalibrate the HS coefficients or parameters [36,39] or modify the equation itself [40,41], aiming to improve its performance.

Reanalysis data or gridded meteorological data are an alternate source of information that can be used to estimate ET_0_ [42,43,44]. It is available on different platforms: National Aeronautics and Space Administration—Prediction of Worldwide Energy Resource (NASA-POWER) [27,45], Global Land Data Assimilation System (GLDAS) [46], Climate Forecast System ver. 2 (CFSv2) [47], North American Land Data Assimilation System (NLDAS) [48], and National Digital Forecast Database (NDFD) [49]. These global or regional platforms provide data with higher spatial and temporal resolution [27]. However, it should be noted that higher spatial resolution does not necessarily imply higher precision [50]. The NASA-POWER platform (NP) is the most widely used to estimate ET_0_ [51,52,53]. NP provides daily information on air temperature, precipitation, relative humidity, radiation, wind direction, and speed; it is free and easily accessible. This information is grouped into three different spatial conditions: for a single point, with time series data available based on registered geographic coordinates chosen by the user; at the regional level, in a time series dataset based on a bounding box of user-determined geographic coordinates; and globally, with climate averages worldwide [54]. Despite the wide availability of information and ease of access, evaluating and validating said NP climate information with in situ weather stations in the area of interest is essential for local bias correction and to improve accuracy [45]. 

This study aims to compare the accuracy of ET_0_ calculated with the BC and HS methods relative to the FAO-56 Penman–Monteith (PM) reference method, with data recorded by an automated weather station (AWS) and temperature data (maximum and minimum) from the NASA–POWER platform (NP), for different calculation periods. 

## 2. Materials and Methods

### 2.1. Study Area and Data Collection

The climatic variables to calculate ET_0_ with empirical equations were recorded with a wireless Davis Vantage Pro 2 Plus AWS (Davis Instruments Company, Hayward, CA, USA); it has a console that allows viewing of all meteorological variables simultaneously [55], with a 30-min update frequency. 

The AWS belongs to Centro Nacional de Investigación Disciplinaria en Relación Agua, Suelo, Planta, Atmósfera (National Center for Disciplinary Research on Water, Soil, Plant, Atmosphere; CENID RASPA) of Instituto Nacional de Investigaciones Forestales, Agrícolas y Pecuarias (National Institute of Forestry, Agricultural, and Livestock Research; INIFAP), within the facilities of an agricultural production unit located at Module XII of the Lagunera Region Irrigation District 017, at 1110 m a.s.l. and coordinates 25°47′00.32″ N, 103°18′46.54″ W (Figure 1). 

Module XII covers an area of 14,276.7 hectares with an elevation range of 1102 to 1114 m [56]. The slope of the area is gentle, at around 0.06%. It is oriented from south to north, with the southern part being the highest, as shown in Figure 2.

Based on data gathered from Series VII of INEGI (2018) [57], it is estimated that the study area is primarily used for irrigated agriculture, with 89.3% of its surface area dedicated to this use. Human settlements make up 8.7% of the area, while the remaining 2% is used for other purposes (Table 1).

The meteorological information used was daily averages for the period between 26 February (Julian day 57) and 9 August (Julian day 221) 2021 (*n* = 165). In the Lagunera Region Irrigation District 017, the main crops of the spring–summer cycle are grown in this period, including forage corn. 

In addition, the meteorological variables were downloaded from the NP climate website (National Aeronautics and Space Administration—Prediction of Worldwide Energy Resource; https://power.larc.nasa.gov, accessed on 5 October 2022). This website collects information from various sources: data recorded on-site, satellite data, wind probes, and assimilated data systems [27]. 

The NNP weather data are based on a single assimilation model named GMAO (Global Modeling and Assimilation Office), starting from the MERRA-2 (Modern Era Retrospective-Analysis for Research and Applications) reanalysis dataset and the GEOS (Goddard Earth Observation System) data processing system [58,59]. Solar radiation is derived from the GEWEX SRB (Global Energy and Water Exchanges Project Surface Radiation Budget) project [60,61]. 

The horizontal resolution of the NP meteorological data source corresponds to a ½° × 5/8° latitude/longitude grid, and the solar data sources come from a 1° × 1° latitude/longitude grid. The current version no longer reassigns data to a common grid; once the data are processed and filed, they are available through the NP service package. The meteorological data is derived from NASA’s GMAO MERRA-2 and GEOS 5.12.4 FP-IT. The NP platform team processes GEOS data daily and combines them with MERRA-2 data, producing daily time series that yield low-latency products usually available in approximately two days (real-time). Energy flow data (solar irradiance, thermal IR, and cloud properties) derive from NASA’s GEWEX SRB Release 4-Integrated Product (R4-IP) file and CERES SYNIdeg and FLASHFlux projects. These data are processed daily and added to the daily time series, issuing products after approximately 4 days, almost in real-time [62].

The main features of the NP system database are shown in Table 2. The AWS is situated near the center of Module XII (Figure 2) and aligns with the center of the NP platform cell. This suggests that one cell encompasses the entire study area’s surface.

### 2.2. ET_0_ Estimation with Empirical Equations 

ET_0_ was estimated through three empirical equations with different information requirements: an equation based on a complete physical model (PM); another on temperature and solar radiation (HS); and the last one on temperature, relative humidity, and wind speed (BC). 

#### 2.2.1. FAO-56 Penman–Monteith Method (ET_0_-PM)

ET_0_ was estimated daily with the FAO-56 Penman–Monteith method using Equation (1) [22,63]; this method is useful for arid, temperate, and tropical zones [19,22].
(1)$${ET}_{0-PM}=\frac{0.408∆\left(R_{n}-G\right)+\gamma \frac{900}{T+273}u_{2}\left(e_{s}-e_{a}\right)}{∆+\gamma \left(1+0.34u_{2}\right)},$$
where $R_{n}$ is the net radiation at the reference crop surface (MJ m^−2^ d^−1^); G is the soil heat flux density (MJ m^−2^ d^−1^); $u_{2}$ is the wind speed at 2 m height (m s^−1^); $e_{s}$ is the saturation vapor pressure (kPa); $e_{a}$ is the actual vapor pressure (kPa); $e_{s}-e_{a}$ is the vapor pressure deficit (kPa); ∆ is the slope of the vapor saturation pressure curve (kPa °C^−1^); T is the mean daily air temperature at 2 m height (°C); and γ is the psychrometric constant (kPa °C^−1^). For daily time intervals, G values are relatively small, and therefore, this term was not included [22]. 

#### 2.2.2. Hargreaves–Samani Method (ET_0_-HS) 

The Hargreaves–Samani method estimates ET_0_ based on temperature data only (Equation (2)) [64]. This equation was developed for semi-arid zones and is useful when solar radiation data are not available; however, as it is based on a few variables, its accuracy should be evaluated at the regional and local levels [65].
(2)$${ET}_{0-HS}=K_{H}\left(T+K_{T}\right)R_{0}\times {\;\left(T_{max}-T_{min}\right)}^{A_{H}},$$
where T is the mean daily air temperature (°C); $R_{0}$ is the extraterrestrial solar radiation (from tables, mm d^−1^); $T_{max}$ is the maximum daily air temperature (°C); $T_{min}$ is the minimum daily air temperature (°C); $K_{H}$ and $K_{T}$ are the empirical calibration parameters; and $A_{H}$ is a Hargreaves’ empirical exponent. This study used the original values proposed by Hargreaves and Samani [64]: $K_{H}\;$= 0.0023, $K_{T}$ = 17.78, and $A_{H}$ = 0.5.

#### 2.2.3. Blaney–Criddle Method (ET_0_-BC) 

The meteorological variables required to apply the Blaney–Criddle method are air temperature, relative humidity, and daytime wind speed (Equation (3) [66].
(3)$${ET}_{0-BC}=a+b\left[p\left(0.46\times T+8.13\right)\right],$$
where a and b are climatic calibration coefficients calculated with Equations (4) and (5), respectively; p is the mean annual percentage of daytime hours (value from tables, decimal); and T is the mean air temperature at 2 m height (°C).
(4)$$a=0.0043\times {RH}_{min}-\frac{n}{N}-1.41,$$
where ${RH}_{MIN}$ is the minimum relative humidity (%); $\frac{n}{N}$ is the ratio between theoretical and actual sunlit hours (value from tables, decimal).
(5)$$b=0.082-0.0041\left({RH}_{min}\right)+1.07\left(\frac{n}{N}\right)+0.066\left(u_{2}\right)-0.006\left({RH}_{min}\right)\left(\frac{n}{N}\right)-0.0006\left({RH}_{min}\right)\left(u_{2}\right),$$
where $u_{2}$ is the mean daily wind speed at 2 m height (m s^−1^). 

### 2.3. Inferential Evaluation Parameters 

Table 3 shows the inferential parameters used for evaluating the empirical equations that estimate ET_0_ (HS and BC), considering the PM method as a reference. Likewise, the climatic information of the NP platform was evaluated when calculating ET_0_ through the reference method (PM_NP_).

In the above equations, $E_{i}$ is the estimated value using the empirical equation; $O_{i}$ is the value obtained with the reference method; $\bar{E}$ is the average of estimated values obtained with the empirical equation; $\bar{O}$ is the average of the values obtained with the reference method; and n is the number of observations. The criteria for interpreting the reliability coefficient are cited in [67].

ET_0_ was estimated in three different ways with empirical equations (daily, mean, and cumulative) using a total of 165 observations. The mean ET_0_ was determined over the 5-day period, as was the cumulative.

## 3. Results and Discussion

The daily ET_0_ calculated by the PM method and with AWS meteorological data (Figure 3) had the peak value (8.8 mm d^−1^) on Julian day 126 (6 May 2021)—on the same day, a wind speed of 5.0 m s^−1^ was recorded, which was higher than the average recorded over the study period (2.2 m s^−1^). On the other hand, the minimum ET_0_ (2.2 mm day^−1^) was recorded on Julian day 192 (11 July 2021)—the day that recorded a solar radiation value of 107.0 W m^−2^, lower than the average for the study period (282.8 W m^−2^). This low radiation was due to atypical conditions: high cloudiness (rainfall of 7.6 mm recorded) and high relative humidity (84.5%). Some authors mention that wind speed and solar radiation are the climatic variables with the most significant influence on ET_0_ estimates in the study area [27]. Other authors reach the same conclusion when performing a sensitivity analysis in other regions [68,69,70].

### 3.1. Comparison of ET_0_ Estimated by Empirical Equations versus the Reference Method

Table 4 shows the monthly and total ET_0_ estimated using the empirical equations and the reference method (PM). Considering the month with the maximum ET_0_ (May, with the HS and PM_NP equations and June with the BC method) and the reference method (PM), HS yielded an ET_0_ value that was 6.6% lower vs. PM; BC, 12.5% lower; and PM_NP, 13.2% higher. However, considering the month with the minimum ET_0_ (February) and the PM method, HS yielded an ET_0_ 12.8% higher vs. PM; BC, 6.8% higher; and PM_NP, 14.2% higher. It is observed that HS and BC underestimate ET_0_ over most of the study period, consistent with the findings reported by some authors for an agroclimatic region similar to the study area [71]. 

However, when considering total ET_0_ (whole study period) and the PM method, HS recorded an ET_0_ value 5.5% lower vs. PM; BC, a value 15.6% lower; and PM_NP, 10.6% higher; therefore, HS was the equation that yielded values closest to the PM method. This is because HS considers temperature and radiation as the main energy sources that promote evapotranspiration [9,27]. 

The results in Table 4 indicate an overestimation of ET_0_ relative to the value obtained with the PM_NP method during the study period. The magnitude of this overestimation is related to the accuracy of each variable and has been reported only when using NP (NASA-POWER) data and the PM method [52,53,72]. 

Table 5 summarizes the relationship between the climatic variables recorded by the AWS and those obtained from the NP platform during the study period, where wind speed (WS) and solar radiation (SR) showed a low and moderate relationship, respectively. This same behavior has been reported by some authors for WS [27,45,73] and SR [74]. By contrast, Tmax and RH recorded a high ratio, and Tmin recorded a very high ratio. Some authors reported similar *R*^2^ values for Tmin, Tmax [58], and RH [27] to those obtained in the present study. WS was the variable that yielded the lowest *R*^2^. This highlights the multiple challenges in determining this variable; these include quality control of the measured data since improving this aspect may return more accurate estimates [75].

Figure 4 depicts the bias in the data recorded by the automated weather station (AWS) relative to NP platform data for the following meteorological variables: temperature (maximum and minimum), relative humidity, solar radiation, and wind speed. It is observed that 44% of the maximum temperature data evaluated (*n* = 165) were virtually unbiased, while 39% of NP data overestimated Tmax by 2.1 °C to 7.5 °C, and the rest of the data (17%) underestimated Tmax by 1.2 °C to 5.5 °C (Figure 4a). Regarding the minimum temperature, 39% of the data evaluated showed bias values close to zero, while 46% of the NP data overestimated Tmin by 1.6 °C to 5.1 °C and the rest (15%) underestimated Tmin by 1.8 °C to 5.3 °C (Figure 4b).

The less biased RH values (values close to zero) were observed in 16% of the evaluated data; the NP platform underestimated RH by 3.0% to 38.8% in 76% of the data, and the rest of the data (7%) overestimated RH by 5.9% to 14.9% (Figure 4c). On the other hand, 45% of the evaluated data showed the minimum differences in solar radiation bias (values between 0 MJ m^−2^ d^−1^ and 1.5 MJ m^−2^ d^−1^), while 36% of the data overestimated radiation by 3.7 MJ m^−2^ d^−1^ to 17.2 MJ m^−2^ d^−1^, and the rest of the data (19%) underestimated radiation by 2.9 MJ m^−2^ d^−1^ to 9.7 MJ m^−2^ d^−1^ (Figure 4d).

Finally, 50% of the WS data showed bias values close to zero. It is also observed that most data (41%) overestimated WS by 1.4 m s^−1^ to 2.9 m s^−1^, and the rest of the data (9%) underestimated WS by 1.2 m s^−1^ to 3.3 m s^−1^ (Figure 4e).

Based on the above, the NP platform tends to overestimate Tmax, Tmin, SR, and WS while it underestimates RH. This same behavior was reported by Jiménez et al., in the study area for Tmin, WS, and RH [27]. 

Estimating the 5-day cumulative ET_0_ improved the values of *R*^2^, *r*, and *c* relative to daily ET_0_ and 5-day mean ET_0_. This behavior is consistent with the one reported by Jiménez et al. [27], who obtained better *R*^2^ and RMSE values when estimating 10-day mean ET_0_ versus daily data. Also, this way of estimating ET_0_ yielded reliability coefficients (*c*) rated as “very good” for BC and PM_NP and “good” for HS. However, PM_NP showed the best *R*^2^, *r*, and *c* values, while HS yielded the best RMSE, PE, MBE, and b (Table 6). The latter parameter returned values close to 1, indicating that the estimated values are statistically similar to observed or reference values [16]. Some authors reported similar RMSE values (1.1 mm d^−1^) when comparing ET_0_ estimated by the HS equation and the PM method on a daily basis [71,76]. However, some authors recorded an RMSE (0.7 mm d^−1^) for 10-day mean data, which is similar to the RMSE value obtained in the present study for 5-day cumulative ET_0_ [27].

When graphically comparing the empirical equations versus the reference method (PM), daily ET_0_ and 5-day mean ET_0_ show a greater variability (Figure 5a,b); the 5-day cumulative ET_0_ returned the best fit, with a lower variability of ET_0_ values between the empirical equations and the PM method (Figure 5c). 

In addition, HS yielded a better fit than the reference method (PM) in the three ways of estimating ET_0_. Other authors have also reported a better fit with the HS equation relative to other methods and have taken PM as a reference for arid and semi-arid regions [16,35,76]. This equation underestimates ET_0_ over most of the study period because the methods based on solar temperature and radiation do not include wind speed [19]. 

### 3.2. Comparison of Estimated ET_0_ with Observed (AWS) versus Estimated (NP) Data 

Table 7 shows the results of the goodness-of-fit tests between ET_0_ calculated by the PM, HS, and BC methods, using maximum and minimum temperature data from the NP platform for the different calculation periods (daily, 5-day mean, and 5-day cumulative). The analyses of variance, with a 95% confidence interval (*p*-value < 0.0001), indicate a significant linear relationship between the PM method and the HS__NP_ and BC__NP_ equations for the three calculation periods. HS__NP_ yielded the best values of inferential parameters versus BC__NP_, except for *R*^2^ and *r*, in the daily ET_0_ estimate. This indicates that HS with NP temperature data is a suitable option for estimating ET_0_ for different periods. In addition, we found that the estimation percent error (PE) is lower than 5% with HS__NP_ for the three ET_0_ calculation periods. In addition, MBE is negative in the three periods, pointing to an underestimation with the HS__NP_ method. Some authors report this same ET_0_ underestimation effect in semi-arid regions during the winter–summer period [27,71].

The 5-day mean ET_0_ estimate recorded the best values in most statistical parameters relative to the mean daily ET_0_. However, the 5-day cumulative ET_0_ estimate recorded the best *r* and *R*^2^ values compared with the other two estimates (daily ET_0_ and 5-day mean ET_0_). These good results are obtained because grouping ET_0_ over five days mitigates the variation in daily temperature associated with precipitation, wind speed, and cloudiness [77].

Figure 6 shows the dispersion of the calibrated HS method (HS__cal_) relative to PM for the different ET_0_ calculation periods, depicting the best data fit obtained using data accumulated over five days.

The comparison of ET_0_ estimates with the HS equation using temperature data recorded by the AWS (HS_AWS_) and from the NP platform (HS_NP_) yielded a high correlation for the three estimates (Figure 7); the 5-day cumulative ET_0_ recorded the highest *R*^2^. These *R*^2^ values indicate the feasibility of estimating ET_0_ using the NP platform’s temperature data and the HS formula [27,71,76]. 

The accuracy of ET_0_ calculated with methods BC and HS was compared versus the FAO-56 Penman–Monteith (PM) reference method using data from an automated weather station (AWS) and the NASA-POWER platform (NP). In this comparison, the HS equation returned the best fit in the different ways of estimating ET_0_: daily, 5-day mean, and 5-day cumulative, with the latter yielding the best fit. 

## 4. Conclusions

The Hargreaves–Samani (HS) method underestimated by 5.5% the reference evapotranspiration (ET_0_) compared to the FAO-56 Penman–Monteith (PM) method considering the total ET_0_ of the study period (26 February to 9 August 2021). This was because the calculation of ET_0_ with the HS equation does not consider wind speed, which influences the evapotranspiration rate sometimes during the year in the study area. Nonetheless, this method is an alternative for calculating ET_0_ in semi-arid regions for which only temperature records are available. 

The HS equation yielded the best estimate relative to the reference method (PM) in the different ways of estimating ET_0_ during the spring–summer crop cycle; the 5-day cumulative ET_0_ showed the best fit. Therefore, this method is suitable for use with remote-sensing data to determine crop evapotranspiration (ET_c_) with 5-day temporal resolution images. It is necessary to conduct testing of HS in various agroclimatic conditions and perform a regional spatial evaluation using data from additional automated weather stations, such as those located within the entire 017 irrigation district. 

The maximum and minimum temperature data from the NASA–POWER (NP) platform was suitable for estimating ET_0_ with the HS equation. This data source is a timely alternative, particularly in semi-arid regions without data from weather stations.

The results showed that NP is a reliable data source for programming medium- and low-frequency irrigation (sprinkler and surface irrigation), which are common in the study area. In addition, they provide spatially comprehensive data, unlike the point values recorded by weather stations, which could be an enormous advantage when studying large regions, such as irrigation districts. 

## Figures and Tables

**Figure 1 sensors-23-07007-f001:**
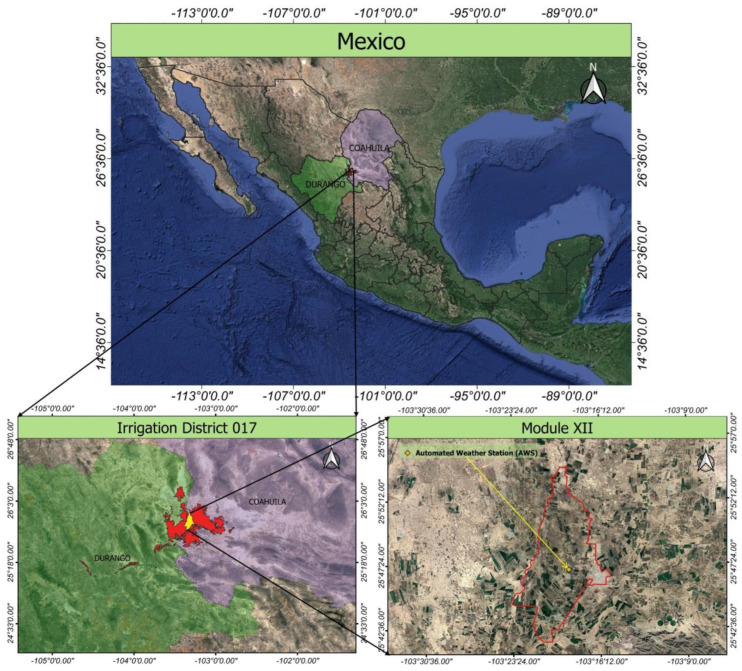
Location of the automated weather station (AWS).

**Figure 2 sensors-23-07007-f002:**
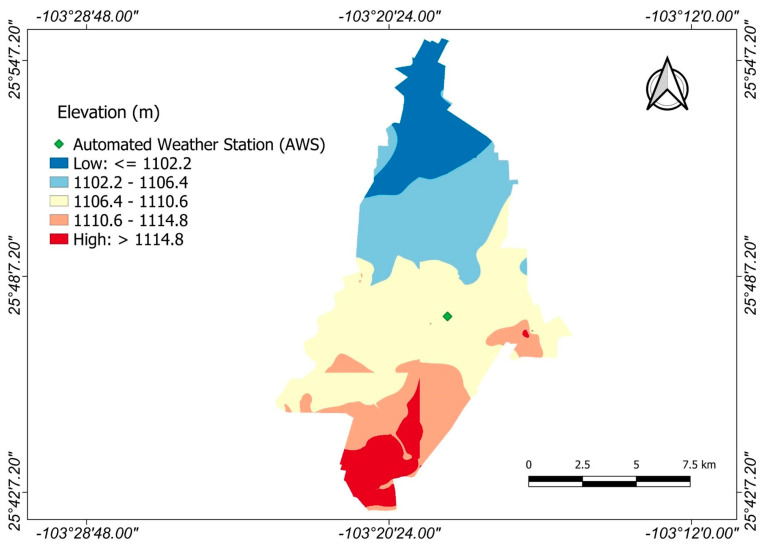
Altitude map of Module XII.

**Figure 3 sensors-23-07007-f003:**
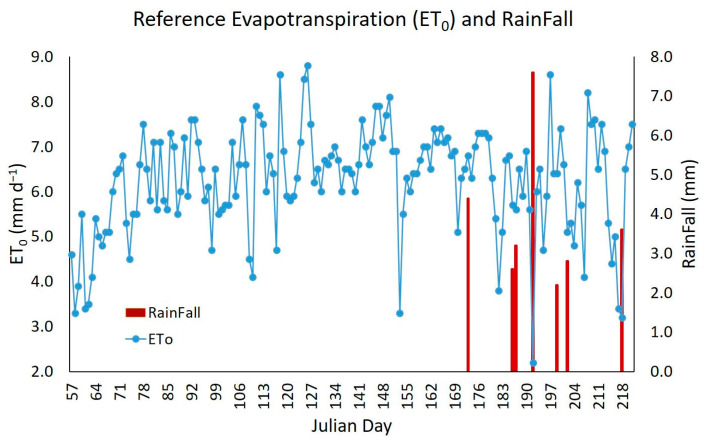
Reference evapotranspiration estimated with the FA0-56 Penman–Monteith method using AWS meteorological data (blue points) and rainfall recorded in the study period (red bars).

**Figure 4 sensors-23-07007-f004:**
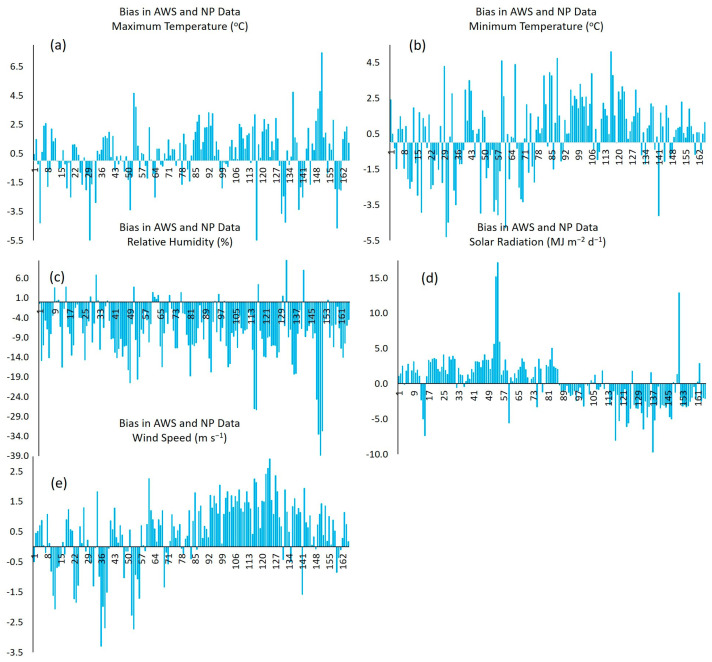
Bias between observed (AWS) and reference (NP) data for the meteorological. (**a**) Bias in AWS and NP Data Tmax. (**b**) Bias in AWS and NP Data Tmin. (**c**) Bias in AWS and NP Data RH. (**d**) Bias in AWS and NP Data SR. (**e**) Bias in AWS and NP Data WS.

**Figure 5 sensors-23-07007-f005:**
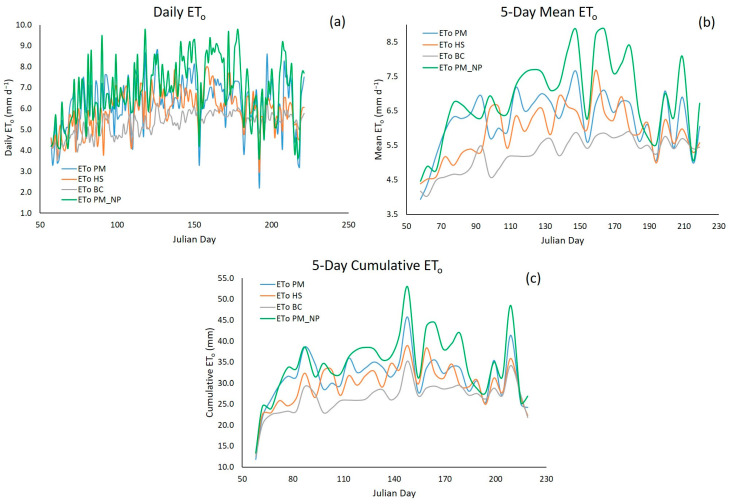
Different ways to estimate ET_0_ using empirical equations and the reference method (PM) during the study period. (**a**) Estimation Daily ET_0_. (**b**) Estimation 5-Day Mean ET_0_. (**c**) Estimation 5-Day Cumulative ET_0_.

**Figure 6 sensors-23-07007-f006:**
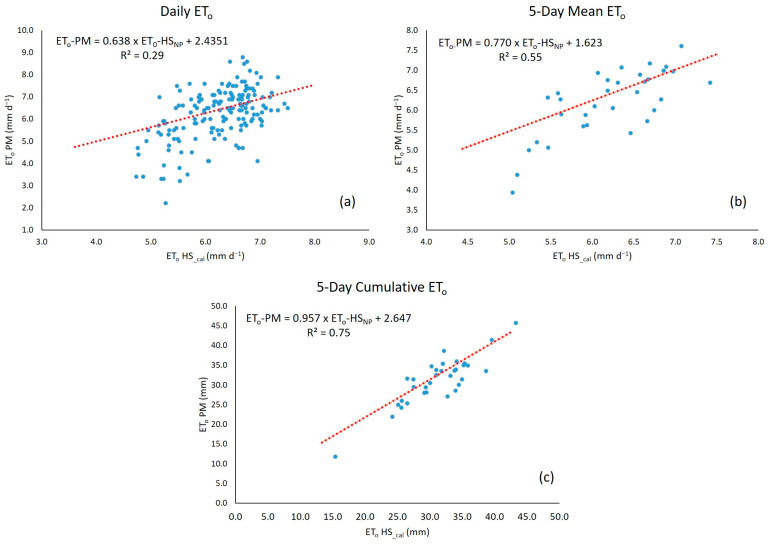
Dispersion plot of the calibrated HS method (HS__NP_) relative to the FAO-56 Penman–Monteith (PM) reference method for the different ET_0_ calculation periods: Daily (**a**), 5-Day Mean (**b**) 5-Day Cumulative (**c**).

**Figure 7 sensors-23-07007-f007:**
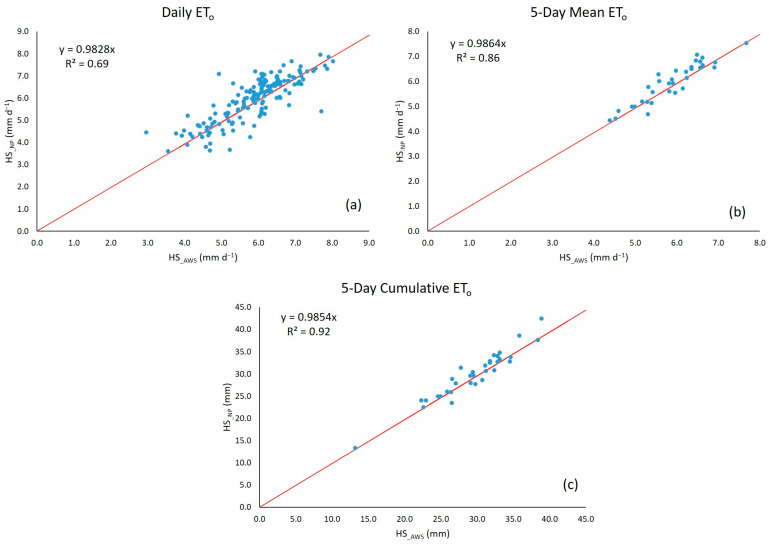
Linear relationship between ET_0_ estimates with the HS equation using temperature data from the AWS and the NP platform for the different ET_0_ calculation periods: Daily (**a**), 5-Day Mean (**b**) 5-Day Cumulative (**c**).

**Table 1 sensors-23-07007-t001:** Land use and vegetation of the study area INEGI-Series VII (2018).

Land Use	Code	Surface (ha)	Coverage (%)
Human Settlements	AH	1240.1	8.69
Barren Land	DV	15.6	0.11
Annual and Semi-permanent Irrigated Agriculture	RAS	10,166.6	71.21
Permanent Irrigated Agriculture	RP	63.0	0.44
Semi-permanent Irrigated Agriculture	RS	2521.3	17.66
Microphyllous Desert Scrub with Secondary Shrub Vegetation	Vsa/MDM	237.1	1.66
Halophilous Xerophytic Vegetation with Secondary Shrub Vegetation	Vsa/VH	32.9	0.23
Total		14,276.7	100.00

**Table 2 sensors-23-07007-t002:** Features of the NASA-POWER (NP) system information.

Parameter	Feature
Data period	1981 to date
Geographic range	Global
Download format	ASCII, CSV, GeoJSON, and NetCDF
Temporal resolution	Daily
Spatial resolution	0.5° × 0.5° (55.56 km × 55.56 km cell) for temperature (T), relative humidity (RH), and wind speed ($u_{2}$). 1.0° × 1.0° for solar radiation and extraterrestrial solar radiation data.
Delayed data availability	Approximately two days for temperature, relative humidity, and wind speed, and five days for solar radiation data.

**Table 3 sensors-23-07007-t003:** Equations and optimal values of inferential parameters.

Parameter	Equation		Optimal Value
Coefficient of Determination ($R^{2}$)	$R^{2}=\frac{{\left[\sum_{i=1}^{n}\left(E_{i}-\bar{E}\right)\left(O_{i}-\bar{O}\right)\right]}^{2}}{\sum_{i=1}^{n}{\left(E_{i}-\bar{E}\right)}^{2}\sum_{i=1}^{n}{\left(O_{i}-\bar{O}\right)}^{2}}$	(6)	1
Root Mean Error (RMSE)	$RMSE=\sqrt{\frac{\sum_{i=1}^{n}{\left(E_{i}-O_{i}\right)}^{2}}{n}}$	(7)	0
Estimate Error Percentage (PE)	$PE=\left|\frac{\bar{E}-\bar{O}}{\bar{O}}\right|*100$	(8)	0
Mean Error Bias (MBE)	$MBE=\frac{\sum_{i=1}^{n}\left(E_{i}-O_{i}\right)}{n}$	(9)	0
Concordance Index (d)	$d=1-\left[\frac{\sum_{i=1}^{n}{\left(E_{i}-O_{i}\right)}^{2}}{\sum_{i=1}^{n}{\left(\left|E_{i}-\bar{O}\right|+\left|O_{i}-\bar{O}\right|\right)}^{2}}\right]$	(10)	1
Correlation coefficient (r)	$r=\frac{\sum_{i=1}^{n}\left(\left|O_{i}-\bar{O}\right|\right)\left(\left|E_{i}-\bar{E}\right|\right)}{\sqrt{\sum_{i=1}^{n}{\left(O_{i}-\bar{O}\right)}^{2}}\sqrt{\sum_{i=1}^{n}{\left(E_{i}-\bar{E}\right)}^{2}}}$	(11)	1
Reliability coefficient (c)	c=r∗d	(12)	1
Regression coefficient (b)	$b=\frac{\sum_{i=1}^{n}O_{i}E_{i}}{\sum_{i=1}^{n}{O_{i}}^{2}}$	(13)	1

**Table 4 sensors-23-07007-t004:** Monthly and total ET_0_ estimated by empirical equations and the reference method (FAO-56 Penman–Monteith) during the study period.

Variable	Evaluation Period: 26 February to 9 August 2021							
	February	March	April	May	June	July	August	Total
	(*n* = 3)	(*n* = 31)	(*n* = 30)	(*n* = 31)	(*n* = 30)	(*n* = 31)	(*n* = 9)	(*n* = 165)
ET_0-PM_ (mm)	11.7	179.0	191.3	214.4	196.7	187.8	49.1	1030.0
ET_0-HS_ (mm)	13.2	154.8	180.8	200.3	195.5	179.6	48.7	972.9
ET_0-BC_ (mm)	12.5	141.3	152.1	171.7	172.2	170.9	48.8	869.5
ET_0-PM_NP_ (mm)	13.4	183.7	204.8	242.7	238.5	203.5	52.3	1138.9

**Table 5 sensors-23-07007-t005:** Relationship between the meteorological variables recorded by the automated weather station (AWS) and obtained from the NP platform during the study period.

Climatic Variables	Coefficient of Determination (*R*^2^)				
	Tmax__NP_	Tmin__NP_	RH__NP_	WS__NP_	SR__NP_
Tmax__AWS_	0.76				
Tmin__AWS_		0.81			
RH__AWS_			0.80		
WS__AWS_				0.27	
SR__AWS_					0.45

Tmax, maximum temperature; Tmin, minimum temperature; RH, relative humidity; WS, wind speed; SR, solar radiation.

**Table 6 sensors-23-07007-t006:** Comparison and inferential parameters to determine the ET_0_ equation that best fits the study area.

Parameter	Methods								
	HS	BC	PM__NP_	HS	BC	PM__NP_	HS	BC	PM__NP_
	Daily ET_0_ (*n* = 165)			5-Day Mean ET_0_ (*n* = 33)			5-Day Cumulative ET_0_ (*n* = 33)		
$R^{2}$ (Dimensionless)	0.29	0.43	0.53	0.44	0.47	0.73	0.69	0.76	0.84
RMSE (mm d^−1^)	1.1	1.3	1.2	0.7	1.1	0.9	3.8	5.8	4.6
PE (%)	5.5	15.6	10.6	5.2	15.3	10.6	5.5	15.6	10.6
MBE (mm d^−1^)	−0.35	−0.97	0.66	−0.32	−0.95	0.66	−1.73	−4.86	3.30
d (Dimensionless)	0.94	0.99	0.98	0.85	1.00	0.94	0.86	1.00	0.94
r (Dimensionless)	0.54	0.65	0.73	0.66	0.69	0.85	0.83	0.87	0.91
c (Dimensionless)	0.51	0.65	0.72	0.56	0.66	0.81	0.71	0.83	0.86
b (Dimensionless)	0.9270	0.8257	1.0994	0.9416	0.8399	1.1073	0.9362	0.8349	1.1075

**Table 7 sensors-23-07007-t007:** Comparison and linear regression coefficients between ET_0_ calculated with the reference method (FAO-56 Penman–Monteith) and HS and BC methods with temperature data from the NP platform.

Parameter	Methods					
	HS__NP_	BC__NP_	HS__NP_	BC__NP_	HS__NP_	BC__NP_
	Daily ET_0_ (*n* = 165)		5-Day Mean ET_0_ (*n* = 33)		5-Day Cumulative ET0 (*n* = 33)	
$R^{2}$ (Dimensionless)	0.29	0.38	0.55	0.45	0.75	0.74
RMSE (mm d^−1^)	1.1	1.3	0.6	1.1	3.3	5.7
PE (%)	4.4	15.0	4.1	14.6	4.4	15.0
MBE (mm d^−1^)	−0.28	−0.93	−0.26	−0.91	−1.38	−4.67
r (Dimensionless)	0.54	0.61	0.74	0.67	0.87	0.86
a (Dimensionless)	2.435	−0.359	1.623	0.732	2.647	−1.700
b (Dimensionless)	0.638	1.244	0.770	1.033	0.957	1.240

## Data Availability

The data are not publicly available because they are currently used in an ongoing thesis.

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
