# Peer review of "Estimation of Reference Evapotranspiration in a Semi-Arid Region of Mexico"

_sensors, 2023, doi:10.3390/s23157007_

Round 1
Reviewer 1 Report
GENERAL COMMENTS
In the study, ET0 was calculated with the Blaney-Criddle, the Hargreaves-Samanithe and the FAO-56 Penman-Monteith methods using information from an automated weather station and the NASA-POWER platform for different periods in a semi-arid region in the North of Mexico. The main aims were to compare the accuracy of ET0 calculated with the different methods and to check whether maximum and minimum temperatures from the NASA-POWER platform were suitable for estimating ET0 using the HS equation.
In general, the paper is well structured, methodologically well described, and the results are clearly presented. The first objective of representing the accuracy of the methods is well presented. The second objective (shown as an essential output in the abstract and the conclusions), whether maximum and minimum temperatures from the NASA-POWER platform were suitable for estimating ET0 using the HS equation, is not presented as an objective in the introduction. Since it is an essential output, it should appear here. Using many different statistical indicators to assess the accuracy of the methods is important, but some indicators, such as r or b, are similar, or their explanatory power is limited. Please check again and use those that can clearly test the quality. The last paragraph in the results and discussion section is a summary that could better fit the conclusions section.
SPECIFIC COMMENTS
Line 139: remove +++
Line 143: (ET0-PM)
Line 145: [14,51]; [11,14]
Line 146: ? is soil heat flux density
Line 147: u2 wind speed at 2 m height
Line 149: T air temperature at 2 m height
Line 152: ET0-HS
Line 157: ? is mean daily air temperature
Line 162: ET0-BC
Line 158-159: ???? is maximum daily air temperature; ???? is minimum daily air temperature
Lines 167: ? is mean air temperature at 2 m height
Line 170: U2 is mean daytime wind speed at 2 m height – is that used in the calculation or daily mean wind speed?
Line 184-185: Translate or omit
Line 215: [1,16]
Line 255: omit [17]
Table 6: B or b for regression coefficient
Table 7: parameter a?
Figure 5, Line 311: be consistent in labelling; table 7 HS_NP, figure 5 HS_cal
Lines 314-315: what is the difference between high correlation and very high? statistically >0.5 is high, change or describe
Author Response
All the observations were answered, which are detailed in the attached document.

Reviewer 2 Report
This manuscript compares the accuracy of ET0 calculated with the Blaney-Criddle (BC) and Hargreaves-Samani (HS) methods versus PM using information from an automated weather station (AWS) and the NASA-POWER platform (NP) for different periods of time. The HS method underestimated by 5.5 % of ET0 compared to the PM method during the period from March to August and yielded the best fit in the different evaluation periods: daily, average, and 5-day cumulative; the latter showed the best values of inferential parameters. The information about maximum and minimum temperatures from the NP platform was suitable for estimating ET0 using the HS equation. This data source is a timely alternative, particularly in semi-arid regions where no data from weather stations are available.
The manuscript is well written, and I have only some minor comments.
Minor comments:
Line 21: and yieldedà but yielded
Line 139: estimated+++à estimated
Line 140: solar temperature and radiationà temperature and solar radiation
Line22, Line 325: what do you mean by the latter?
Author Response

(The authors gave the same response as above.)

Reviewer 3 Report
Reconsider after major revision

Moderate changes in sentences
Author Response

(The authors gave the same response as above.)

Round 2
Reviewer 3 Report
Please check the attachment.

Moderate
Author Response
Dear reviewer, I've responded quickly to your observations, which helped improve the manuscript we submitted for publication. We could not attend to some of these observations as they are outside the scope of the manuscript due to time constraints and lack of information, but we assure you that we did our best.
Kind regards

Round 3
Reviewer 3 Report
Accept in present form
Minor